# Enzymatic Activity and Microbial Diversity of Sod-Podzolic Soil Microbiota Using 16S rRNA Amplicon Sequencing following Antibiotic Exposure

**DOI:** 10.3390/antibiotics10080970

**Published:** 2021-08-12

**Authors:** Tatiana Trifonova, Anastasia Kosmacheva, Alexander Sprygin, Svetlana Chesnokova, Olga Byadovskaya

**Affiliations:** 1Department of Soil Geography, M.V. Lomonosov Moscow State University, Moscow 119991, Russia; 2Department of Biology and Ecology, Vladimir State University Named after Alexander and Nikolay Stoletovs, Vladimir 600000, Russia; hijadelaluna@mail.ru (A.K.); chesnokova.chemist@mail.ru (S.C.); 3FGBI “Federal Centre for Animal Health” (FGBI “ARRIAH”), Vladimir 600901, Russia; spriginav@mail.ru (A.S.); bjadovskaya@arriah.ru (O.B.)

**Keywords:** benzylpenicillin, oxytetracycline, tylosin, microbiome, bacterial community

## Abstract

Antibiotic contamination of the environment negatively affects soil fertility by disrupting natural microbial communities. Currently, the study of the effect of antibacterial drugs on soils typical in Russia, which are of great importance for agriculture, is insufficient. Despite a rapid increase in the number of metagenomic studies, this article is the first publication devoted to the microbial diversity of sod-podzolic soil and its relationship with enzymatic activity. In the present study, we use 16S rRNA metagenomic sequencing to analyze microbiota dynamics and to examine soil enzymatic activities after antibiotic treatment with benzylpenicillin, oxytetracycline, and tylosin. We found that, following treatment, urease activity was reduced regardless of the antibiotic used while nitrification activity showed no statistically significant changes (*p* > 0.05). Oxytetracycline and tylosin produced no effect on catalase activity but benzylpenicillin caused an increase. Benzylpenicillin and oxytetracycline increased cellulolytic activity whereas tylosin had no significant effect (*p* > 0.05). Microbiome profiling through 16S rRNA gene sequencing demonstrated antibiotic administration and exhibited no significant impact on bacterial abundance and species diversity (*p* > 0.05), thus pointing to the resilience of the soil microbial community. Oxytetracycline, benzylpenicillin, and tylosin are likely to negatively affect the enzymatic profiles in sod-podzolic soil but with a negligible influence on the bacterial composition.

## 1. Introduction

Animal husbandry intensification is associated not only with a rise in economic indicators but also with a number of negative impacts on the environment such as the use of medicines against animal infectious diseases, including various spectrum antibiotics. The antibacterial drugs introduced are mainly excreted from the animal body with metabolic products, which are partially used for fertilizing agricultural land and greenhouses [1,2,3,4,5]. Thus, the antimicrobials introduced into the environment threaten natural microbial communities due to their ability to affect soil microorganisms, to change the total biomass, to affect the relative number of different groups, and to change soil enzyme activity. As a result, soil fertility suffers. In this regard, the study of antibiotics as soil ecosystem pollutants has garnered considerable attention. The assessment of enzymatic activity changes and microbial diversity using metagenomics or 16S rRNA gene sequencing under the impact of antibiotic drugs provides significant diagnostic indicators in such research [6,7,8,9,10,11,12,13,14,15,16].

A number of the published studies have been devoted to the impact of antibiotics on soil enzymatic activity changes [5,7,8,9,10,11,12,13]. A urease activity change was detected for oxytetracycline introduction, thus affecting the nitrogen availability in soil [9]. The exposure to different dosages of tylosin, benzylpenicillin, and oxytetracycline caused catalase activity changes in ordinary chernozem, brown forest, and dark chestnut soils depending on both the soil type and the drug properties and concentrations [11,12,13]. The stimulating and inhibitory effects of tylosin, ampicillin, oxytetracycline, and benzylpenicillin (50–700 mg/kg) on the nitrification, urease, catalase, and cellulolytic activities of various soil types in model laboratory experiments have been described in several published studies [17,18,19,20].

The antibiotic ability to cause both enzyme activity decrease and increase is usually associated with changes in the bacterial community number and diversity. However, this issue is poorly studied nowadays, thus complicating our comprehension of the reasons for fertile soil changes in response to anthropogenic pressure [7,14,15,16]. Oxytetracycline introduction has caused diversity changes in the soil microbiota, pig manure, and wheat straw [14,15]. The effect of 2000 mcg/g tylosin demonstrated a temporary change in the structure of the soil bacterial community and a tendency to recover alongside the incubation period increase [16].

A number of publications have used 16S rRNA amplicon sequencing to research the effects of tylosin on the soil microbial community [16], soil bacterial communities under the influence of manure and tetracycline [21], the effects of slurry from sulfadiazine- and difloxacin-medicated pigs on the structural diversity of microorganisms in bulk and rhizosphere soil [22], and the soil bacterial community response to sulfadiazine in the soil–root zone [23]. 

Thus, the study of the replacement of several dominant bacterial species with others under the influence of antibacterial drugs can deepen the understanding of the ecological characteristics of the studied soil. The results obtained can be used in the development of approaches to preserve the natural structure of microbial communities and to ensure soil fertility. Consequently, the research objective was to study the microbial community dynamics and to identify their relation to the changes in catalase, urease, nitrification, and cellulolytic activities in the sod-podzolic soil cultivated in response to benzylpenicillin, oxytetracycline, and tylosin. As sod-podzolic soil is the main arable soil type of the Vladimir region and, alongside podzolic soil, accounts for up to 12.3% of the agricultural land and 14.7% of the arable land in Russia in general, its use in this research is of great practical importance [24].

## 2. Results

### 2.1. Agrochemical Soil Indicators

The agrochemical soil indicators are presented in Table 1. According to the results obtained, the studied soil is characterized by a weak acidity, medium humus content, low nitrate nitrogen content, and the absence of ammoniacal nitrogen, which is typical for this type of soil. A very high content of mobile forms of phosphorus and potassium has been established, which indicates a large amount of fertilizers applied. The object of this study is sod-podzolic medium loamy soil, which is the main type of arable soil in the Vladimir region and accounts for up to 12.3% of the agricultural land and 14.7% of the arable land in Russia [24].

### 2.2. Determination of the Soil Enzymatic Activity

The results of the soil enzymatic activity are shown in Figure 1. The amount of urease activity in the control was 0.951 ± 0.031 mg NH_3_/10 g per 24 h. All three antibiotics significantly reduced the urease activity of the soil. The Fisher test values were F = 81.70 (*p* = 0.00) for benzylpenicillin, F = 62.98 (*p* = 0.00) for oxytetracycline, and F = 119.32 (*p* = 0.00) for tylosin. The nitrifying activity in the control was 996.65 ± 35.84 mg NO_3_^−^/kg. The introduction of antibiotics did not affect the nitrification. The Fisher test values were F = 0.84 (*p* = 0.41) for benzylpenicillin, F = 1.36 (*p* = 0.31) for oxytetracycline, and F = 1.85 (*p* = 0.25) for tylosin. Cellulolytic activity in the control was 47.35 ± 2.11%. A statistically significant increase was caused by benzylpenicillin (F = 61.10, *p* = 0.00) and oxytetracycline (F = 11.31, *p* = 0.02); tylosin had no effect on this indicator (F = 1.60, *p* = 0.25). The catalase activity in the control was 5.06 ± 0.06 mL O_2_/min per 1 g. Benzylpenicillin increased the enzyme activity (F = 69.11, *p* = 0.00); no statistically significant effect of oxytetracycline (F = 1.43, *p* = 0.30) and tylosin (F = 0.09, *p* = 0.78) was found.

### 2.3. Metagenomic Sequencing

The data processing resulted in 132,272 reads identified (30,000 reads per sample). After quality control and filtration, 3205, 3042, 3693, and 4500 nucleotide sequences were detected in samples 1, 3, 4, and 5, respectively, forming 201 operational taxonomic units (OTUs), 40 of which were unique (i.e., an OTU without any available information in the database). The numbers of OTUs per sample, as well as the Shannon, Simpson, and Chao1 indices, are shown in Table 2 and Appendix A.

To quantify the proportion of the soil microbiome in each sample, the ratio of the detected OTUs to the Chao1 index was applied. The Chao1 index evaluates the theoretical number of species/OTUs in the studied microbiome. The population coverage amounted to over 95% of the theoretically calculated true microbiome diversity. The statistical analysis of the Chao1 index demonstrated the absence of significant diversity in the theoretical number of species/OTUs in the studied microbiome.

The Shannon diversity index in the four samples fluctuated from 4.49 to 5.21. This index characterizes the microbiome bacterial diversity. The higher the Shannon index, the greater the species diversity of the community. The results obtained indicated a high species diversity of communities in all studied samples. The statistical analysis demonstrated the absence of a significant variability in the taxonomic structure.

The Simpson index indicates the dominance of certain community members. When analyzing the samples, the dominance index varied from 0.88 to 0.95. The obtained values of the Simpson index demonstrated a high diversity of prokaryotic communities in all studied samples. The statistical analysis indicated a uniform distribution without the predominance of any representatives.

The Feit phylogenetic diversity index was 7.7–8.0. This index is the sum of the lengths of the branches of the phylogenetic tree uniting all species of the studied sample and reflects the length of the evolutionary history in a given set of species. The statistical analysis demonstrated no significant changes in the microbial community structure in terms of the phylogenetic relation.

The Appendix A, which characterizes the dependence of the diversity of the microbiome in the form of detected OTUs on the selective effort (the number of sequenced nucleotide sequences). Appendix A proves that, at 2000 reads per sample, the number of detected OTUs almost reached a plateau, which indicates the adequacy of the sampling effort in reflecting the theoretical diversity. The comparison of the change in the abundance of microbiota representatives by the grouped OTU are shown in Figure 2. Thus, Figure 2 shows the groups of bacteria in the control (sample 1) and those that survived after antibiotic treatment (samples 3–5).

The metagenomic 16S rRNA analysis showed that the samples contained Gram-negative bacteria such as the cyanobacteria genus *Tychonema* (o. *Nostocales*) and the bacteria genus *Pseudomonas*. The family *Chitinophagaceae* (o. *Chitinophagales*), containing genus *Arthrobacter* (o. *Micrococcales*), genus *Massilia* (o. *Burkholderiales*), genus *Lysobacter*, and genus *Arenimonas* (o. *Xanthomonadales*), was also present. Representatives of the archaea family *Nitrososphaeraceae* (o. *Nitrososphaerales*) were also present to the least extent. The dominant bacterium in samples 1, 3, and 4 was cyanobacterium genus *Tychonema* (o. *Nostocales*) whereas the dominant bacterium in sample 5 was genus *Pseudomonas*.

Figure 3 presents a thermal map of the difference between the bacterial abundance among the tested samples. For compactness, the OTUs are grouped according to the taxonomic order.

Figure 3 shows that the abundance of certain representatives of the studied microbiomes varied depending on the antibacterial drug. Each specimen type possessed its own taxonomic abundance composition, phylogenetically related to each other. Despite the visible differences in Figure 2 and Figure 3 in the structure of the microbiomes in the studied samples, no statistically significant changes occurred in the microbiome composition (*p* > 0.05).

The bacteria o. *Nostocales* (cyanobacteria), o. *Frankiales*, o. *Gaielleles*, o. *Nitrososphaerales* (archaea), and o. *Opitutales* dominated in the control. Sample 3 was characterized by the two groups consisting of bacteria *Pyrinomonadales*, *Solibacteriales*, *Gemmatimonadales*, and *Microtrichiales* and *Chitinobacteriales*, *Streptomycetales*, *Sphingomonadales*, *Rhizobiales*, *Micrococcales*, and *Pseudonocardiales*. The dominant bacteria for group 4 were *Sphingobacteriales*, *Blastocatellales*, *Caulobacterales*, *Xanthomonadales*, *Chitinophagales*, and *Steroidobacteriales*. The dominant composition for Group 5 consisted of *Pseudomonadales, Gammaproteobacteria, Pedosphaerales, Bacillales,* and *Azospirillales*.

Figure 4 shows the principal component analysis (PCoA) data of the control and antibiotic-treated samples.

The figure exhibits that the microbiome cluster of sample 5 (exposed to tylosin) differed from the control and samples 3 (benzylpenicillin) and 4 (oxytetracycline). It is important to note that no statistically significant changes occurred in the microbiome structure (*p* > 0.05).

## 3. Discussion

We believe that this research is the first metagenomic study conducted in the Russian Federation on the bacterial diversity of sod-podzolic soil in response to the impact of antibacterial drugs. This work is a continuation of previous studies. However, if—in previous works—the effect of antibiotics on enzymatic activity was studied, then this work focuses on the prokaryotic structure and its relationship with the enzymatic processes under the influence of antibiotics [17,18,19,20]. Several papers have been published concerning only the assessment of enzymatic activity dynamics without studying the microbiome and without determining the observed activity and, through it, soil fertility [11,12,13].

In this study, we evaluated enzymatic activity and changes in the soil microbiome in response to antibiotics.

The urease activity inhibition, when exposed to 200 mg/kg of oxytetracycline, was consistent with the results of a similar study [9] and the tylosin effect corresponded with previously published data dealing with this antibiotic effect on a different soil type [17,18]. Thus, all three antibiotics, differing in their properties and spectrum of action, were capable of suppressing the urease activity of the studied soil at the 200 mg/kg concentration. The urease activity inhibition is likely associated with changes in the archaea content of the *Nitrososphaeraceae* family, in which representatives can use urea as a substrate [25]. However, the content decrease in these microorganisms did not significantly affect the soil microbiome composition due to their low content (Figure 2).

The stimulating impact of benzylpenicillin detected on catalase activity can be explained by the bacterial increase with catalase activity regarding genus *Arthrobacter* (Figure 2) [26]. The obtained results were identical to similar data concerning a catalase activity increase under tylosin influence on chernozem and brown forest soils during the incubation of the contaminated samples and in dark chestnut soil at 1–100 mg/kg of the antibiotic in 3 days, which the authors associated with active enzyme production in response to the drug introduction [13].

The increase in cellulolytic activity under the influence of benzylpenicillin and oxytetracycline is likely explained by fungal enzymatic activity, which is not sensitive to these drugs, thus causing a fungal biomass increase in contrast to the bacterial biomass decrease under the influence of the antibiotics introduced. Moreover, the fungi killed by bacteria can be used as an additional energy source [16]. The insignificant inhibitory impact of tylosin on cellulolytic activity (*p* > 0.05) might depend on the additional substances in the drug composition enhancing its antiseptic effect in relation to both bacteria and fungi. Moreover, it was presumably associated with the established changes in the bacterial community composition regarding the bacterial predominance of the genus *Pseudomonas* (Figure 2) and the difference between this microbiome cluster and other samples (Figure 4). However, these indicators were also not statistically significant (*p* > 0.05).

The lack of an effect of the studied antibiotics on the soil nitrification activity can be explained by the spectrum of activity of the antibiotics. The nitrification process was carried out by Gram-negative bacteria and archaea [10]. The nitrification organisms in the studied soil were represented by the archaea of the *Nitrososphaeraceae* family although their contribution was negligible. The decrease in the nitrate ion content in the soil under tylosin impact can be explained by the increase in the number of bacteria from the genus *Pseudomonas*, which is involved in denitrification processes, in the microbiota composition [27]. Benzylpenicillin is effective against Gram-positive bacteria and, therefore, it cannot suppress nitrification microorganisms. The spectrum of oxytetracycline and tylosin action is wide; thus, it causes a reduction in nitrification activity. However, the dispersion analysis data revealed statistically significant differences in the nitrification activity neither in the control nor under the influence of these drugs (*p* > 0.05).

The bacterial community structures in the studied samples did not differ statistically significantly in terms of species diversity, the dominance of any taxon, or phylogenetic diversity (*p* > 0.05), thus demonstrating a certain resistance of soil communities in response to antibiotic exposure. The principle taxonomic groups of bacteria practically did not change (*p* > 0.05) but their abundance in the sample changed. *Cyanobacteria* (o. *Nostocales*) and genus *Massilia* (family *Burkholderiaceae*) predominated in these groups. Insignificant amounts of archaea were available in the control (sample 1 (untreated soil)) whereas they were not found in the experimental samples after antibiotic treatment. With a dominance of the genus *Pseudomonas* in sample 5 (Figure 2) in comparison with that in samples 1, 3, and 4, the difference in microbiome cluster 5 (Figure 4) depended on the wide spectrum of tylosin action and the additional substances in the composition of the drug used (in particular, propylene glycol and benzyl alcohol), which possess antiseptic properties [28]. However, the difference between these indicators was statistically negligible (*p* > 0.05).

## 4. Materials and Methods

### 4.1. Soil

The sod-podzolic mid-loamy soil was selected in compliance with GOST 17.4.4.02-84 from an agricultural site in the Vladimir region, Russia (56°14′47″ N, 40°34′51″ E), at 0–20 cm depth. The samples were taken using the envelope method from five points at the 1 × 1 m site and thoroughly mixed. The sampled soil was air-dried at room temperature and sifted through a 2 mm sieve. Four homogeneously similar samples were taken from it, with one serving as the control (sample 1). Samples 3, 4, and 5 were treated by adding aqueous solutions of the antibiotics benzylpenicillin, oxytetracycline, and tylosin, respectively, at concentrations corresponding with 200 mg/kg of soil. Distilled water was used to prepare the antibiotic solutions. Before DNA isolation, the soil samples were stored in sterile plastic sealed containers at −10 °C for three months.

### 4.2. Determination of the Soil Agrochemical Characteristics

The salt extract acidity (pH_KCl_) was determined using 1 M solution of potassium chloride in the soil and a solution ratio of 1:2.5 following the potentiometric method.

The organic matter content was determined following the photoelectrocolorimetric method using a photometer KFK-3-01 based on the organic matter oxidation with a potassium dichromate solution in sulfuric acid and the subsequent determination of trivalent chromium equivalent to the organic matter content.

The physical clay content (granulometric composition) was determined using the mass content of various sizes expressed as a percentage relative to the mass of the dry soil sample taken for the analysis.

Nitrates were extracted from the soil with an aluminum-potassium alum solution with 1% mass fraction at the ratio of soil sample mass to the solution volume of 1:2.5 and with the subsequent determination of the nitrates in the extract using an ion-selective electrode (Expert-001, Econix-Expert, Rumyantsevo, Russia).

Exchange ammonium was extracted from the soil with a potassium chloride solution to obtain a colored indophenol compound formed via the interaction of ammonium with hypochlorite and sodium salicylate in an alkaline medium and the subsequent photometry of the colored solution using a photometer KFK-3-01.

The mobile potassium (K_2_O) and phosphorus (P_2_O_5_) compounds were extracted from the soil using a hydrochloric acid solution of 0.2 mol/dm^3^ molar concentration and determined using a KFK-3-01 photometer.

### 4.3. Antibiotics

Tylosin, referred to as the macrolides group, was used in the drug formulation of the injectable solution with an active substance concentration of 200 mg/cm^3^. The antibiotic produces a bacteriostatic effect and has the potential for time-dependent bactericidal action.

Oxytetracycline, referred to as the tetracyclines group, was used in the oxytetracycline hydrochloride composition. The drug is characterized by a wide spectrum of activity against Gram-positive and Gram-negative bacteria. In most cases, it exhibits a bacteriostatic effect.

Benzylpenicillin, referred to as the β-lactam antibiotic group, was used in the composition of benzylpenicillin sodium salt. It exhibits bactericidal and bacteriostatic effects and is efficacious against Gram-positive organisms [29].

The process antibiotic solutions were prepared by diluting the initial preparations in distilled water. The studied concentrations (200 mg/kg of soil) were selected based on the published data [9,10,11,12,13] and our research results concerning changes in the soil enzymatic activity under the impact of a wide range of antibiotic concentrations [17,18,19,20].

### 4.4. Enzymatic Activity Determination

The soil enzymatic activity was determined in model laboratory experiments.

The soil cellulolytic activity was determined using the modified Christensen application method based on the calculation of decomposed cellulose. One cotton cloth piece of size 3 × 4 cm^2^, serving as a source of fiber, previously kept in a drying cabinet at 105 °C for 2 h, and weighed to 0.0001 g accuracy, was placed at the bottom of each sterile Petri dish. Afterwards, 50 g of the test soil, moistened to 60% of the total moisture capacity and contaminated with antibiotic solutions, was added into each Petri dish. The Petri dishes were weighed, placed into the thermostat, and incubated at 27 °C for 30 days. The soil was moistened daily with distilled water. In 30 days, the remaining cloth was removed from the Petri dishes, cleaned from the soil, dried at 105 °C, and weighed to 0.0001 g accuracy. The difference in the cloth mass (%) decomposed during the experiment served as the soil cellulolytic activity indicator [30].

The catalase activity was determined by applying the gasometric method to determine the rate of hydrogen peroxide decomposition in the soil by the volume of oxygen released during the reaction. For this purpose, the 1 g soil samples, moistened up to 60% of the total moisture capacity and contaminated with antibiotics, were exposed at a temperature of 22 °C for 5 days. The control was the original soil without any drug introduction. Furthermore, a 1 g soil lot was placed in a flask, 0.5 g calcium carbonate was added and attached to the system for the gasometric determination of the catalase activity, 5 mL of a 3% hydrogen peroxide solution was added, and the released oxygen volume was measured for 1 min at constant stirring in an orbital shaker [31].

The soil urease activity was determined by applying the photocolorimetric method by counting the amount of ammonia formed during the hydrolysis of the introduced urea. For this purpose, 1 g of soil was placed in a flask and 5 mL of a 3% urea solution and 1 drop of toluene were added. The flasks were sealed with cork stoppers, shaken, and exposed in a thermostat at 30 °C for 24 h. After the incubation period, 15 mL of 1.0 n. KCl solution was added into each flask and stirred in an orbital shaker for 5 min. Afterwards, 10 mL of the soil extract was transferred into a centrifuge tube and centrifuged for 5 min at 3000× rpm. An amount of 1 mL of the centrifugate was placed in a 50 mL measuring flask and 30 mL of distilled water, 2 mL of a 30% solution of potassium-sodium tartaric acid, and 2 mL of a Nessler chemical agent were added. After each addition, the solution was stirred, reaching the desired volume by adding distilled water, and was thoroughly mixed and colorimetrated in a KFK-3-01 photometer in 30 mm wide cuvettes with a blue light filter at a 400 nm wavelength. Urease activity was expressed in milligrams of NH_3_ per 10 g of soil per day [32]. 

To determine the nitrification activity, 100 g of soil was placed into a sterile conical flask and moistened with distilled water up to 65% of the total moisture capacity by adding the solutions of the appropriate antibiotics. A total of 0.1 g of ammonium sulfate and 0.2 g of calcium carbonate were added into each suspension, covered with a cotton plug, and incubated in a Sanyo MLR-351 climate chamber for 30 days at a constant temperature of 27 °C in the dark. The soil moisture was maintained by the addition of distilled water weekly to reach its initial level. To assess the nitrification activity, the nitrate ion content in the samples was measured after the incubation period: the soil from each flask was placed into Petri dishes, air-dried at 40 °C, and thoroughly mixed; 20 g was taken from each sample and 50 mL of a 1% aluminum-potassium alum solution was added and mixed by the orbital shaker for 20 min. The nitrate ion content in the samples was determined and measured by applying the potentiometric method (Expert-001, Econix-Expert, Rumyantsevo, Russia). To assess the antibiotic effect on the soil nitrification activity, the results of the nitrate ion content in the sample that did not contain antibiotics (value 0 in the graphs) were compared with the value when the corresponding antibiotics were added [32].

The catalase, urease, and nitrification activities were determined in triple replication and cellulolytic activity was determined four-fold. The initial soil without the introduction of antibiotics was considered to be the control.

### 4.5. Statistics Analysis

The results were statistically processed using Statistica 7.0 software. To statistically analyze the effect of the antibiotic concentration on enzymatic activity, a single-factor analysis of variance was used including a Fisher F-test determination (significance level: *p* < 0.05). The standard error values were indicated as the error. The graphs were plotted in Microsoft Excel.

Alpha- and beta-analyses of the biodiversity among the OTU frequencies were performed using a CLC Genomics Workbench (Qiagen, Hilden, Germany) with default parameters. The alpha-diversity was evaluated using the Shannon diversity index, providing estimations of the diversity species; the phylogenetic diversity index, demonstrating the taxa affinity degree represented in the compared communities; the Chao1 index, estimating the total number of taxa in a community; and the Simpson index, demonstrating the probability of two individuals being randomly selected from the indefinitely large community of different taxa. The Kruskal-Wallis test was used to assess the statistical significance among the groups.

To estimate the beta-diversity, the “weighted unifrac” method was applied, allowing for the estimation of the percentage of similarities/differences among the samples, taking into account the phylogenetic information. The results were presented using the multivariate statistics methods of the PCoA analysis. The data visualization was performed in the CLC Genomics Workbench program.

### 4.6. DNA Preparation for Sequencing

A set of NucleoSpin Soil chemicals (Macherey-Nagel, Düren, Germany) was used to isolate the DNA from the soil samples. The v3–v4 region of the 16S rRNA gene was used as the target amplification site [33].

A PCR with an F515 direct primer and an R806 reverse primer was performed in a 15 µL reaction mixture containing a 0.5–1 activity unit of Q5^®^ High-Fidelity DNA Polymerase (NEB, Ipswich, MA, USA), 5 pcM of direct and reverse primers, 10 ng of a DNA matrix, and 2 nM of each dNTP (LifeTechnologies, Foster City, CA, USA). The mixture was denatured at 94 °C for 1 min, followed by 35 cycles: 94 °C for 30 s, 50 °C for 30 s, and 72 °C for 30 s. The final elongation occurred at 72 °C for 3 min. The PCR products were purified using AMPureXP (BeckmanCoulter, Chaska, MN USA). Further library preparations were carried out in accordance with the manufacturer’s MiSeq Reagent Kit Preparation Guide (Illumina San Diego, CA, USA).

### 4.7. Sequencing

The amplicon library analysis of the ribosomal operon fragments by high-performance 16S rRNA gene sequencing was performed using an Illumina MiSeq system (Illumina, USA) and a MiSeq ^®^ ReagentKit v3 (600 cycle) reagent kit with a double-row reading (2 × 300 n). The data received after the sample sequencing were processed using CLC Genomics “Workbench” software. Clustering into the OTU and annotation were performed according to the SILVA 16S v132 99% database, a part of the Microbial Genomics Module. The reading quality analysis, pair-end sequence integration, and chimera removal were performed using the Microbial Genomics Module with default parameters. The OTU was grouped according to the level of taxonomic similarity at a threshold of 97% and the total occurrence reliability was at least 10 OTUs per analyzed group of samples.

The results were obtained using the equipment from CCP “Genomic Technologies, Proteomics, and Cell Biology” of FGBNU VNIISKHM.

## 5. Conclusions

The combination of 16S rRNA amplicon sequencing methods with classic methods for studying the enzymatic activity of soils is an important way to solve problems in the ecology of soil microorganisms concerning the relationship between the structure of microbial communities and ongoing biogeochemical processes. Despite a rapid growth in the number of metagenomic studies, this article is the first publication devoted to the microbial diversity of sod-podzolic soil and its relationship with enzymatic activity. As the soil microbiome is characterized by a richness of species diversity, the continuation of such studies is required, including a more complete study of the effects of a wide range of antibacterial drugs comprising not only the prokaryotic pool but also soil fungi. Metagenomic profiling by 16S rRNA revealed the dominance of Gram-negative bacteria in the genus *Tychonema* in the soil without antibiotic exposure, which remained under the treatment with benzylpenicillin and oxytetracycline. However, when treated with tylosin, the microbiome composition shifted towards the dominance of the *Pseudomonas* genus. Despite the detected resistance of the microbial community of the studied soil in response to benzylpenicillin, oxytetracycline, and tylosin, changes in the soil enzymatic processes were observed depending both on the enzymatic activity type and on the drugs used. Thus, antibiotic introduction might negatively impact the enzymatic processes of soils without a significant influence on the microbial community structure.

## Figures and Tables

**Figure 1 antibiotics-10-00970-f001:**
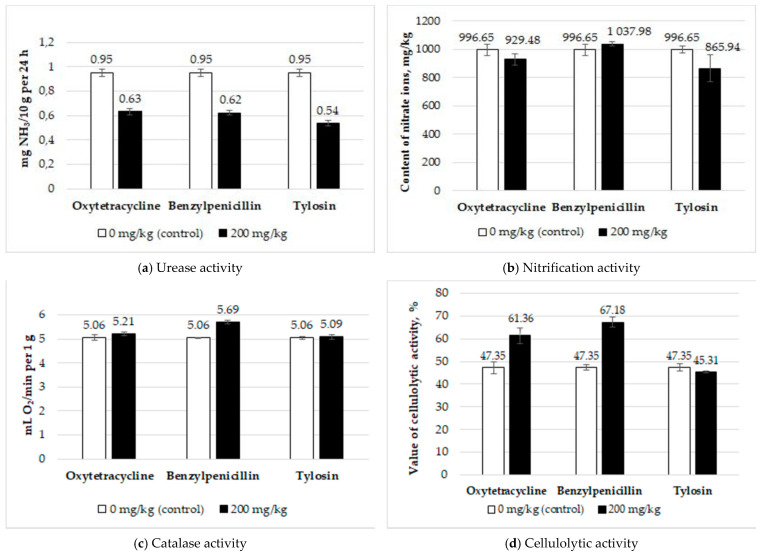
The amount of soil enzymatic activity in the control (0 mg/kg) and with antibiotics (200 mg/kg): (**a**) Urease activity, (**b**) Nitrification activity, (**c**) Catalase activity, (**d**) Cellulolytic activity.

**Figure 2 antibiotics-10-00970-f002:**
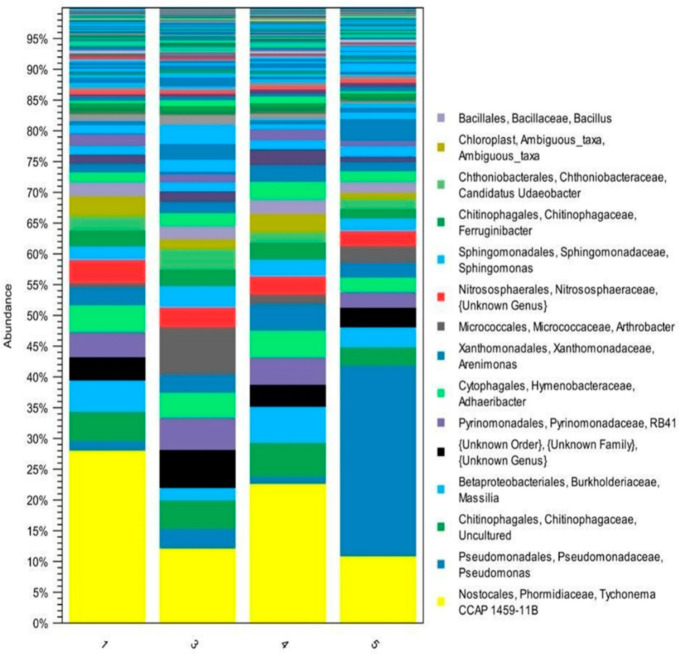
Graphical comparison of the bacterial abundance at the genus level for the studied samples.

**Figure 3 antibiotics-10-00970-f003:**
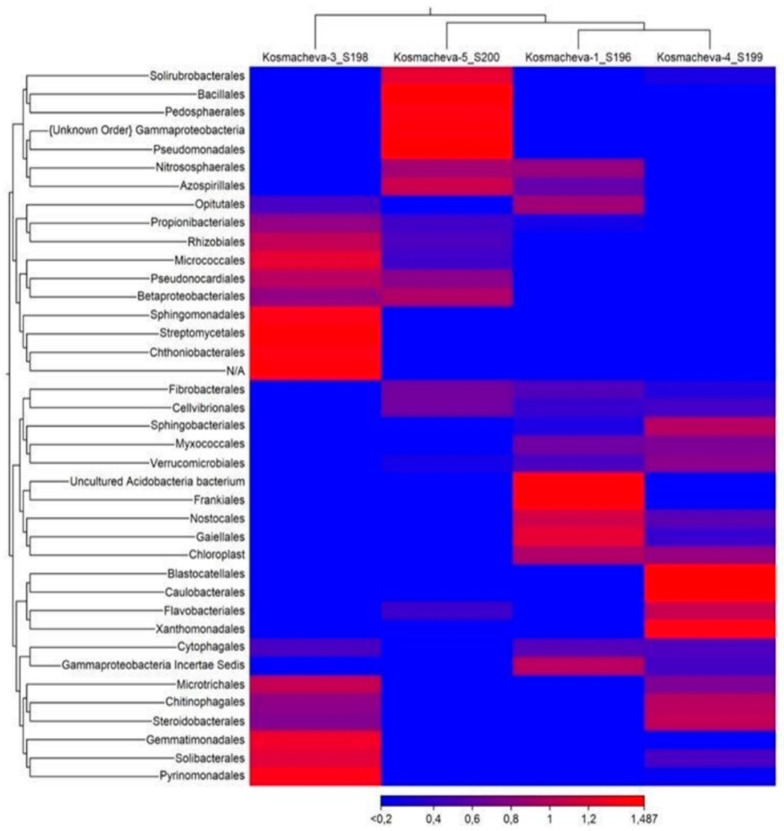
Thermal map of the bacteria abundance dynamics at the order level. The color bar indicates OTU relative abundances expressed as Z-scores.

**Figure 4 antibiotics-10-00970-f004:**
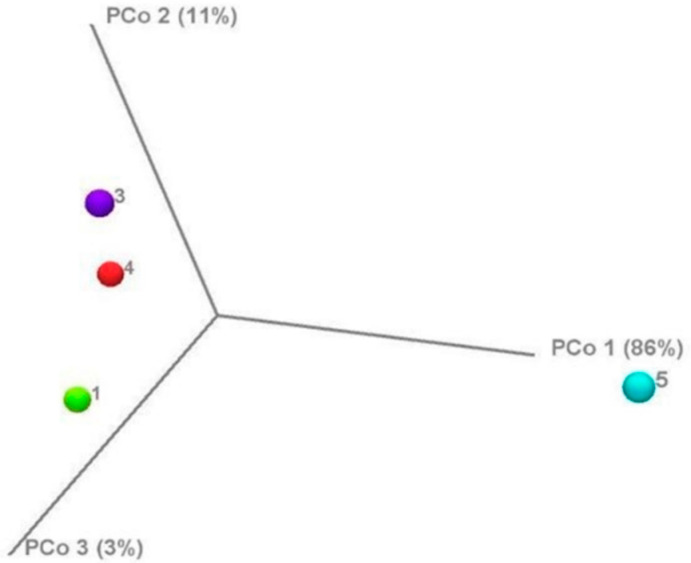
Principal component analysis (PCoA) showing the differences between the untreated (control) and antibiotic-treated soil samples determined using the weighted UniFrac diversity metric. Each colored symbol corresponds with an individual sample (1: untreated soil sample (control), 3: soil treated with benzylpenicillin, 4: soil treated with oxytetracycline, and 5: soil treated with tylosin). The variation represented by each axis (PC1, PC2, or PC3) is shown in parentheses.

**Table 1 antibiotics-10-00970-t001:** Agrochemical soil indicators.

Indicator, Unit	Value
Acidity, pH_KCl_	5.56 ± 0.20
P_2_O_5_ mobile, mg/kg	436.5 ± 87.30
K_2_O mobile, mg/kg	275.6 ± 41.34
Organic matter, %	2.36 ± 0.47
N ammonium, mg/kg	0
N nitrates, mg/kg	10.1 ± 1.52
Physical clay, %	30.2

**Table 2 antibiotics-10-00970-t002:** Alpha-diversity metrics.

Sample	OTU	Chao1 Index	Shannon Index	Simpson Index	Feit Phylogenetic Diversity
1	186	189.8	4.74	0.90	7.91
3	191	199	5.21	0.95	7.77
4	194	202	4.90	0.92	8.01
5	197	201.9	4.49	0.88	7.92

## Data Availability

All reported findings are available within the manuscript.

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
