# Peer review of "Enzymatic Activity and Microbial Diversity of Sod-Podzolic Soil Microbiota Using 16S rRNA Amplicon Sequencing following Antibiotic Exposure"

_antibiotics, 2021, doi:10.3390/antibiotics10080970_

Round 1

Reviewer 1 Report

The manuscript “Enzymatic activity and metagenomic profiling by 16S rRNA of sod-podzolic soil microbiota at antibiotics exposure” by Tatiana Trifonova, Anastasia Kosmacheva, Alexander Sprygin, Svetlana Chesnokova and Olga Byadovskaya describes the results of microflora dynamics studied using 16S rRNA and evaluates changes in enzyme activity in soil in response to antibiotics.

I suggest the following adjustments:

Chapter 2.1 and 2.2  - add a brief description of the results and then insert a table and figure;

Lines 75-85 - the authors report statistical significance but data are missing;

Figure 2 - complete the legend, 1 -5 samples;

Lines 225, 227, 298 - add name, resp. device manufacturer.

Author Response

Dear Reviewer!

Thank you for reviewing our article and important recommendations.

We have made adjustments:

Chapter 2.1 and 2.2: A brief description of the results has been added.

Lines 75-85: Expanded description of results, indicated statistical data.

Figure 2: completed the legend.

Lines 225, 227, 298 - added name, device manufacturer.

Best regards,

Authors.

Reviewer 2 Report

The ms "Enzymatic activity and metagenomic profiling by 16SrRNA of sod-podzolic soil microbiota at antibiotic exposure" by Trifonova and collaborators is an interesting paper describing the effect of antibiotics used to prevent and to treat animal infections in fertilized soils. The results are interesting but the ms needs a serious improvment in the english. 

Comments and suggestions:

  • Title, I recommend to use "microbial diversity" instead of "metagenomic profiling".
  • Line 19 and along all the ms. "Check" is not an appropriated term, use "controls" instead.
  • Line 21, eliminate "of"
  • Line 32, use drugs instead of medecines
  • Line 35, use introduction instead of ingress
  • Lines 61-62, rephrase
  • Line 65, use "was" instead of "is"
  • Lines69-74, too many figures and tables without any text. Improve presentation
  • Line 83, "absolute terms"? It is not clear what it means
  • Table 2, keep one or to decimals in the "Feit phylogenetic diversity"
  • Figure 2, include it as Supplementary Information, remove it from the text
  • Line 115, "small amounts" is not a microbiological term
  • Line 141, explain the symbols used in the Figure 5, change "check" by "controls" as mentioned before
  • I recommend a thorough revision by a english speaking microbiologist.

Author Response

Dear Reviewer!

Thank you for reviewing our article and important recommendations.

We have made adjustments:

  • Title, I: rused "microbial diversity" instead of "metagenomic profiling".
  • Line 19: used "controls" instead "Check".
  • Line 21: eliminated "of".
  • Line 32: used drugs instead of medecines.
  • Line 35: used introduction instead of ingress.
  • Lines 61-62: rephrased.
  • Line 65: used "was" instead of "is".
  • Lines69-74: Improved presentation.
  • Line 83: Line 115 – edited.
  • Table 2: edited decimals in the "Feit phylogenetic diversity".
  • Figure 2: included it as Supplementary Information, removed it from the text.
  • Line 141: explained the symbols used in the Figure 5, changed "check" by "controls".

Best regards,

Authors.

Reviewer 3 Report

The Manuscript presents interesting solutions and results. However, it should still be improved.

  1. Please be sure that a manuscript thoroughly establishes how this work is fundamentally novel. Specific comparisons should be made to previously published materials that have a similar purpose. Please present a strong case for how this work is a major advance. This needs to be done in the manuscript itself, not just in the response to review comments. This is a very important point in terms of which I will further consider the manuscript.
  2. Please be sure that the abstract and the conclusions section not only summarize the key findings of the work but also explain the specific ways in which this work fundamentally advances the field relative to prior literature.
  3. Words used in the title should not be repeated in the keywords.
  4. The introduction lacks information on the types of mechanisms of antibiotics on the metabolic activity of soil microorganisms.
  5. The discussion section should be supplemented with more recent publications on the research subject.
  6. Indicate the possible risks of such research. Add your recommendations for future research.
  7. Please standardize the print and subscriptsin the figures. The print is illegible
  8. Make sure the references are added correctly according to the journal's instructions.
  9. The language correctness should be verified by a native speaker.

Research carried out by the author seems to be important  to the development and enhancement of existing information on this subject. The paper can be accepted for publication after the aforementioned corrections have been made.

Author Response

Dear Reviewer!

Thank you for reviewing our article and important recommendations.

We have made adjustments:

  1. Please be sure that a manuscript thoroughly establishes how this work is fundamentally novel. Specific comparisons should be made to previously published materials that have a similar purpose. Please present a strong case for how this work is a major advance. This needs to be done in the manuscript itself, not just in the response to review comments. This is a very important point in terms of which I will further consider the manuscript.
  2. Please be sure that the abstract and the conclusions section not only summarize the key findings of the work but also explain the specific ways in which this work fundamentally advances the field relative to prior literature.
  3. Words used in the title should not be repeated in the keywords.
  4. The introduction lacks information on the types of mechanisms of antibiotics on the metabolic activity of soil microorganisms.
  5. The discussion section should be supplemented with more recent publications on the research subject.
  6. Indicate the possible risks of such research. Add your recommendations for future research.
  7. Please standardize the print and subscriptsin the figures. The print is illegible
  8. Make sure the references are added correctly according to the journal's instructions.
  9. The language correctness should be verified by a native speaker.

Best regards,

Authors.